

# Identifying global patterns of stochasticity and nonlinearity in the Earth System

Fernando Arizmendi[1], Marcelo Barreiro[1], and Cristina Masoller[2]

[1]Instituto de Física, Facultad de Ciencias, Universidad de la República, Iguá 4225, Montevideo, Uruguay
[2]Departament de Física, Universitat Politecnica de Catalunya, 08222 Terrassa, Barcelona, Spain

*Correspondence to:* C. Masoller (cristina.masoller@upc.edu)

**Abstract.** We demonstrate that two simple measures of time series analysis are able to capture different dynamical and statistical properties of large-scale atmospheric phenomena. We consider two surface air temperature (SAT) datasets, covering a spatial grid of points over the Earth surface (NCEP CDAS1 and ERA Interim reanalysis). In each location we analyze i) the distance between the lagged SAT time series and the insolation (i.e., the local top-of-atmosphere incoming solar radiation), and ii) the Shannon entropy computed from the probability distribution function (pdf) of SAT values. The distance quantifies the similarity between the lagged SAT waveform and the isolation waveform, while the entropy, as defined in information theory, measures the degree of disorder or uncertainty of each time series, which we refer to as stochasticity: the entropy captures the shape of the SAT pdf and is maximum when the pdf is uniform. With the distance measure we uncover well-defined spatial patterns formed by regions with similar SAT response to solar forcing, while with the entropy measure, we uncover regions that have SAT pdf of similar shape. The entropy analysis also allows identifying the geographical regions in which SAT time series has extreme values (i.e., values which are extreme in the local statistics), because the long-tail shape of the pdf is captured as low entropy values. We uncover significant differences between the NCEP CDAS1 and ERA Interim datasets in specific geographical regions, which are due to the presence of extreme values in one dataset but not in the other. In this way, the entropy maps are a valuable tool for anomaly detection and model inter-comparisons.

## 1 Introduction

Large-scale climate phenomena have attracted great interest in the last decades, as improving the understanding of the complex nature of climate interactions is crucial for advancing long-term forecasts. For climate research, a wide range of models of different levels of complexity are available (Dijkstra, 2013), and to perform model inter-comparisons and quantify uncertainties, a lot of research is nowadays devoted to the development of appropriated climate data analysis tools (Mudelsee, 2014).

A data-driven approach that has been proven to be valuable is based on complex networks. Within this approach, a climate network is defined over a regular grid of geographical locations (nodes) (Tsonis et al., 2006; Tsonis and Swanson, 2008; Yamasaki et al., 2008; Donges et al., 2009), and the links between pairs of nodes are defined by performing bi-variate statistical similarity analysis between the time series of climate variables recorded at the nodes.



Different measures have been used to define the links, such as the cross-correlation, the mutual information, the conditional

mutual entropy, Granger causality, etc. (Barreiro et al., 2011; Martin et al., 2013; Tantet and Dijkstra, 2014; Hlinka et al., 2014; Fountalis et al., 2015; Deza et al., 2015; Tirabassi et al., 2015; Tirabassi and Masoller, 2016), and similar, or different, network
structures have been unveiled, depending on the measure employed and the significance confidence level used.

Important challenges for inferring the real underlying connectivity of the climate system include the role of the Sun in

forcing the whole climate system, and the role of "noise", i.e., climate variability with different time-scale with respect to the "signal" being considered. For example, when the analysis is focused on phenomena at inter-annual or longer time-scales,
weather systems concentrated at 3-7 days can be regarded as noise.

Within the network approach, if the time series in two nodes have similar characteristics (due to a similar local response

to common solar forcing and/or similar variability), these regions are likely to appear a "linked", but there can be no genuine underlying relation between the climate variables in the two regions. This can be expected because of the physical processes
that govern our climate (ocean and atmospheric processes, solar forcing, vegetation, human activity, etc.), act in a similar way in distant regions, have similar effects, and therefore, distant regions can have similar climatic properties. Examples include
tropical rainforests, dry and arid regions, maritime regions, etc. Such similar local properties of the time series in two regions might be reflected as network links, depending on the similarity measure used to construct the network. Recently, by using two
network construction methods intended to detect climate similarities rather than interactions, Tirabassi and Masoller (2016) uncovered the set of regional communities formed by nodes located in different hemispheres, which have similar properties of
SAT time series. On the other hand, if the network links are intended to represent real climatic interactions, climatic similarities can obscure the interpretation of the links inferred, because, depending on the similarity measure used, distant nodes with
similar climate can appear a "linked" without having underlying interaction processes between them. It is therefore important to perform uni-variate time series statistical analysis to yield light into the interpretation of the links inferred from bi-variate
time series analysis.

Here we characterize the properties of surface air temperature (SAT) time series (monthly NCEP CDAS1 and ERA Interim

reanalysis), focusing in the response to annual solar forcing and in the "noisy" nature of SAT variability. We chose the SAT field because it has been commonly used to define climate networks.
To quantify the response to solar forcing, we use a distance measure to compare the time-lagged SAT waveform with the isolation waveform (i.e., the local top-of-atmosphere incoming solar radiation); to quantify the "noisy" nature of SAT variability
(to which we refer to as stochasticity), we use a well-known information theory measure: the Shannon entropy (Shannon, 1948). Shannon entropy quantifies the degree of disorder or uncertainty, and is maximum when the probability distribution function
(pdf) of SAT values is uniform.

By using these two measures we aim at answering the following questions: Can we identify geographical regions with

similar response to solar forcing? Where are the regions with strongest distortion of the response? Can we identify regions that have similar SAT stochasticity? Where are the regions with strongest SAT stochasticity?
In the literature, a wide range of distance and entropy measures are available to investigate time series (Beck and Schögl, 1995; Bandt and Pompe, 2002). Here, to quantify the distance between two time series, the insolation, $x$, and the SAT, $y$, we



use a simple procedure. Because we are interested in assessing the similarity of the waveforms, the two time series are first

normalized to zero mean and unit variance; then, the SAT time series is shifted an appropriated number of months, $\tau$, with

respect to the insolation; then, we compute the distance as $d = \sum_t |x(t) - y(t)|$, which is known as *taxicab* metric. In the

*Supplementary Information* we demonstrate that similar results are obtained with the Euclidean metric, $\sum_t [x(t) - y(t)]^2$. A

potentially important drawback of the distance measure used here is that it is non-zero for general forms of linear convolutions

[i.e., it returns an non-zero value for the distance between $x$ and $y$, if they are linearly related as $y(t) = \int_0^t g(t - t')x(t')dt'$]. In

spite of this drawback, our results demonstrate that, at least for the analysis of monthly SAT time series, the taxicab distance

yields meaningful results, when the isolation, $x$, and the SAT, $y$, are compared by using an appropriately chosen lag, $\tau$.

Regarding stochasticity measures, Shannon entropy provides information about the *shape* of SAT pdf, and another suitable

measure can be the *width* of SAT pdf, i.e., the standard deviation of the pdf. A drawback of both measures is that the their

values are not affected if the data is shuffled randomly, because randomly shuffling the data values in the time series does

not modify either the shape or the width of the pdf. An alternative approach is based in nonlinear symbolic analysis (Kantz

and Schreiber, 2003). In this approach the time series is first transformed into a sequence of symbols, and then, the entropy

is computed from the probability distribution of the symbols. In this case, depending on the specific rule employed to define

the symbols, the symbolic entropy will capture different properties of the ordering of the values in the time series, and will

give a different result when the data values are shuffled randomly. It will be interesting, for future work, to compare the results

presented here with those obtained by using different distance and entropy measures.

## 2    Datasets and measures

We consider monthly mean SAT data from two reanalysis data sets: NCEP CDAS1 (Kistler et al., 1996) and ERA Interim

(ERA Interim, 2000). The spatial resolution is $2.5^o/1.5^o$ and cover the time-period [1949-2011]/[1979-2013] respectively. The

NCEP CDAS1 reanalysis has $N = 10224$ time series of length $L = 756$ while the ERA Interim, $N = 28562$ and $L = 408$. The

insolation at the top of the atmosphere is calculated following Berger (1978), as a function of day of year and latitude. Then,

monthly averaged values for every latitude are calculated.

As discussed in the Introduction, to analyze the local response to solar forcing we compute the distance between SAT time

series in grid point $i$, $\{y_i(t)\}$, with the insolation time series in the same grid point, $\{x_i(t)\}$, where $i \in [1, N]$ and $t \in [1, L]$. As

we are interested in assessing the similarity of the shape of $x$ and $y$ waveforms, we first normalize the two time series to zero

mean and unit variance. Then, we compute their distance as

$$d_i = \sum_t |x_i(t) - y_i(t + \tau_i)|. \tag{1}$$

Here $\tau_i$ is a lag that takes into account inertia and/or memory effects, and needs to be appropriately chosen. This lag is

expected to be more important in the oceans in comparison with land masses, because of the larger heat capacity of water. A

natural choice is the value of $\tau_i$ that minimizes the distance between the insolation and the climatology (the averaged monthly

SAT); similar results were found when considering the raw SAT time series instead. We search the minimum of the distance





considering $\tau_i$ values in the interval of [0-4] months because of the lack of physical mechanisms that could result in a longer

delayed response of the climatology to the insolation. The robustness of the results with respect to the maximum lag is shown

in the Supplementary Information, Fig. S1.

To quantify the stochasticity of SAT time series, we calculate Shannon entropy of both, raw SAT and SAT anomalies

(computed by removing the mean annual cycle from each time series). The entropy is normalized to the maximum entropy, of

the uniform distribution:

$$H_i = -\sum_{n=1}^{M} p_n^i \log p_n^i / \log M. \tag{2}$$

Here $\{p_n^i\}, n \in [1, M]$ is the probability distribution of the values in the $i$th time series ($i \in [1, N]$) and $M$ is the number of

bins. $M$ is the same for all time series within a reanalysis database, but is adjusted in each database to take into account the

different length of the time series: we use $M = 20$ for ERA Interim and $M = 40$ for NCEP CDAS1. This gives similar average

data points per bin ($\sim 20$).

The bin size is determined by the local extreme values in each time series, i.e., $dy_i = (\max\{y_i\} - \min\{y_i\})/M$. While

this choice, at first sight, might seem contradictory with performing "inter-site comparisons", it allows to resolve with adequate

precision the shape of the pdf in each site. If global extreme values are considered to define a uniform bin size, a good resolution

of the shape of the pdf in each site would not be possible, because of the very large SAT and SATA variations in certain sites

(e.g., close to the poles), as compared to the small variations in other sites (close to the equator). Therefore, if a uniform bin

size is used, in the sites where SAT and SATA variations are small, the shape of the pdf will be poorly resolved because most

of the bins will be empty, while a few number of bins will contain most of the data points.

## 3   Results

Figure 1 presents the results of the forcing-response analysis. Figure 1(a) displays the map of $d_i$ values computed without

lagging the SAT time series with respect to the insolation, while Fig. 1(b) displays the same but lagging the SAT time series

a value $\tau_i \in [0, 4]$. Figure 1(c) displays the map of $\tau_i$ values. These maps were obtained from the analysis of the ERA Interim

dataset, and similar results were obtained from the NCEP CDAS1 dataset (see Fig. S3 in the *Supplementary Information*).

In Fig. 1(a) we observe large distance values in the tropics with coherent spatial structures over the oceans, in the cold

tongues and areas associated with easterly trades and upwelling processes. Over the continents $d_i$ values are considerably

smaller. There are exceptions, as in the Amazon and the African rainforest, which show high $d_i$ values.

When the SAT time series is shifted, Fig. 1(b), the largest $d_i$ values appear over the tropical rainforests of Africa and South

America. These are regions dominated by monsoons and can be expected to have large $d_i$ values because during the summer

rainy season, when the insolation has its highest values, the solar energy is used mainly for evaporation instead of for heating.

In the oceans there are coherent spatial structures, with high $d_i$ values, in regions which tend to coincide with regions of deep

convection in the Atlantic, Pacific and Indian oceans, including the Intertropical Convergence Zone (ITCZ). In these regions

the SAT and rainfall are strongly coupled so that relatively small changes in SAT gradients modulate and shift the ITCZ. In



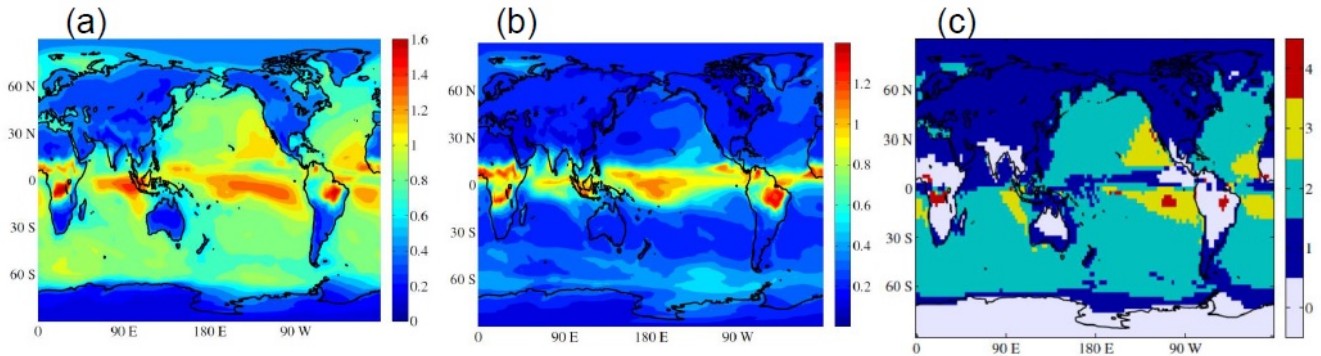

**Figure 1.** (a) Map of distances $d_i$ calculated from Eq. (1) when the forcing (insolation) and the response (SAT) are not shifted ($\tau_i = 0$). (b) Map of distances, when the forcing and the response are shifted $\tau_i$, with $\tau_i \in [0, 4]$. (c) Map of $\tau_i$ values. Data from ERA Interim reanalysis.

particular, air-sea coupling in the eastern basins induce oceanic cold tongues which together with the continental geometry

maintain warm waters and the ITCZ to the north of the equator. Thus, the high distance values in this region can be interpreted

as due to the strong air-sea coupling. Outside the 10°S-10°N band, the higher $d_i$ values over the eastern subtropical north

Pacific, as compared to the western basin, can be due to the influence of the semi-permanent anticyclone and associated stratus

clouds which shield solar radiation. High latitude oceans (southern Ocean, Labrador sea, Greenland sea) also have relatively

large $d_i$ values, which can be explained by the existence of seasonal sea ice in the regions.

The map of $\tau_i$ values shown in Fig. 1(c) uncovers a rather symmetric behavior between both hemispheres. Overall, extra-

tropical land masses have a lag of about 1 month, while extratropical oceans present a lag of 2 months, in agreement with

McKinnon et al. (2013). In tropical oceanic areas $\tau_i$ displays values in the [0-4] range, with values close to 0 and 1 in the ITCZ

region and $\tau_i = 3$ in the eastern basins dominated by stratus clouds. Two continental regions, the African tropical rainforest

and the Amazon rainforest, have a lag of 4 months, which could be due to the fact that in these regions SAT is colder in the

summer compared to the spring because of intense rainfall.

Figure 2 presents the analysis of the stochasticity of SAT variability. Figures 2(a) and 2(b) display the entropy calculated

from ERA Interim reanalysis: from SAT time-series in panel (a) and from SAT anomalies in panel (b). The spatial patterns

uncovered in Fig. 2(a) resemble those in the map of distance values displayed in Fig. 1(b), computed from the same reanalysis

after shifting the SAT time-series.

We note however that there is an opposite trend: regions with high $H_i$ have low $d_i$, and vice-versa. To quantify and visualize

this opposite trend we present in Fig. 3 a scatter plot of both quantities. The linear correlation coefficient between $H_i$ and $d_i$

is $-0.45$. In this plot we also show that the tropical and extratropical regions have different characteristics in the $d$-$H$ plane;

as the inter-quartile range shows, tropical regions (displayed with squares) have larger $d$ and smaller $H$ in comparison to the

extra-tropics (displayed with circles). We also note that the distance values allow for a good separation of the two regions, a

fact that is not captured by the entropy.



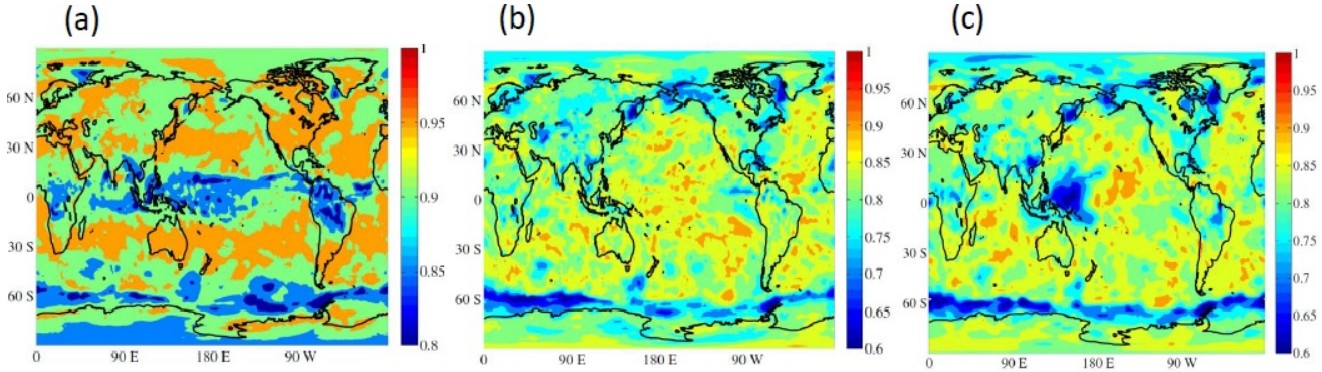

**Figure 2.** (a) Map of Shannon entropy of SAT time series and (b) map of Shannon entropy of SAT anomalies, calculated from ERA Interim reanalysis. The spatial structures in panel (a) are similar to those observed in Fig. 1(b). (c) Map of Shannon entropy of SAT anomalies calculated from NCEP CDAS1 reanalysis. Comparing panels (b) and (c), we note a well-defined region in the western Pacific where the two data sets display significant differences.

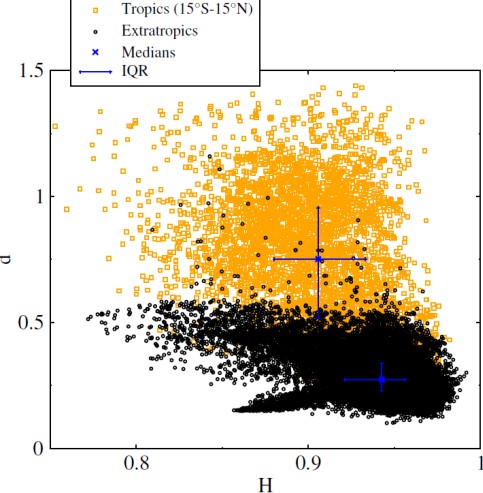

**Figure 3.** Entropy-distance scatter plot. The tropics ($15°$S-$15°$N) and extratropics ($90°$S-$15°$S, $15°$N-$90°$N) are represented in orange squares and black circles, respectively. The blue crosses represent the respective medians and the blue lines, their correspondent inter-quartile ranges.



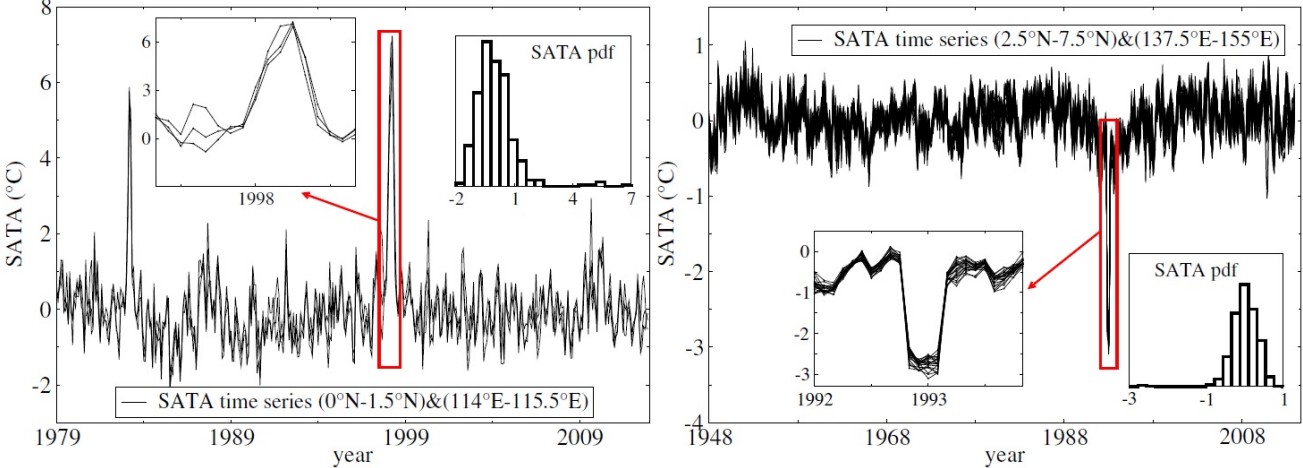

**Figure 4.** SAT anomaly time series displaying extreme values in ERA Interim (*left*) and in NCEP CDAS1 reanalysis (*right*). The left (right) panel shows the temporal evolution of the 3 (21) grid points within the indicated domain.

Comparing the entropy map obtained from SAT, Fig. 2(a), with that obtained from SAT anomalies, Fig. 2(b), we note that in the anomaly map only spatial patterns associated with sea ice remain, while the structures seen in the tropics in Fig. 2(a), disappear in Fig. 2(b).

The persistence of the patterns associated to sea ice in the SAT anomalies map can be explained by the fact that in these regions there is a strong seasonality in SAT variance that is not removed with the mean seasonal cycle. In these regions, in the winter the temperature can decrease considerably as the ice caps insulate the atmosphere from the ocean. In consequence, SAT pdf and SAT anomalies pdf show a long tail in low temperature values, which is captured by smaller entropy values.

To check whether this is a robust result, we computed the entropy of SAT anomalies using NCEP CDAS 1 reanalysis. The resulting map is presented in Fig. 2(c) (see also Fig. S4): it is very similar to that obtained from ERA Interim except in the western tropical Pacific where NCEP data shows lower entropy. An inspection of SAT time-series in this region reveals the existence of extreme values (outliers) which render the pdf to be peaked in a narrow interval, which in turn results in low entropy values. In ERA Interim there are also outliers, but, because they cover a very small area, their low entropy values cannot be resolved in the entropy map shown in Fig. 2(b). The time series of SAT anomalies in these regions, displaying extreme values, are presented in Fig. 4.

## 4  Discussion

As discussed in the Introduction, a motivation of our work is to analyze the local properties of SAT time series, to yield light into the interpretation of the links inferred by bi-variate network analysis. In this section we contrast our findings with related network studies.



Tirabassi and Masoller (2013) performed a detailed analysis of the effects of lag-times in networks constructed from monthly SAT reanalysis. The maps of the lag-times between SAT time series in different regions, (Tirabassi and Masoller, 2013, Fig. 2), for the three regions considered (Australia, El Niño basin and Mongolia) have a structure that is remarkably similar to the map presented in Fig. 1(c) here; however, Tirabassi and Masoller used a different procedure to compute the maps: the lag was determined as the value where the cross-correlation between two SAT time series in two geographical regions is maximum. In other words, the lag between any two regions was determined such as to superpose the two seasonal cycles. Here, the lag is determined by minimizing the distance between SAT time series and the isolation, in the same region. The two approaches give several consistent observations. For example,

i) in (Tirabassi and Masoller, 2013, Fig. 2, right panel) the large red area in the North Hemisphere represents the geographical regions where the seasonal cycle is in-phase with that in Mongolia; comparing with Fig. 1(c) here we note that these are regions with a one month lag between the seasonal cycle and the isolation;

ii) in (Tirabassi and Masoller, 2013, Fig. 2, left panel) the large red area in the Southern Hemisphere represents the regions where the seasonal cycle is in-phase with that in Australia; comparing with Fig. 1(c) we note that in those regions, the lag between the seasonal cycle and the isolation is 0 or 1 month;

iii) in (Tirabassi and Masoller, 2013, Fig. 2, center panel) the red area in the equator represents the regions where the seasonal cycle is in-phase with that in El Niño basin; comparing with Fig. 1(c) we note that several well-defined structures appear in both maps, in particular the regions in yellow in Fig. 1(c), where the lag between the seasonal cycle and the isolation is 3 months, correspond to regions either in red or in light blue in (Tirabassi and Masoller, 2013, Fig. 2, center panel).

The fact that there are well-defined mutual lags between SAT time series in different regions was then used in Tirabassi and Masoller (2016) to infer *climate communities*, defined as regions that share similar properties of SAT time series. The map of the communities obtained, which are the regions which have a synchronous seasonal cycle, Fig. 1 in Tirabassi and Masoller (2016), has also several similar features to the map presented in Fig. 1(c). For example, the large communities formed by the oceans in the north and in the south hemispheres, represented with orange and blue in (Tirabassi and Masoller, 2016, Fig. 1), have in our map a 2 month lag between SAT and insolation; two communities in the equatorial Pacific and Atlantic oceans shown with violet and green in (Tirabassi and Masoller, 2016, Fig. 1) have the same lag, 3 months, between SAT and insolation.

We can also relate our findings with the work by Hlinka et al. (2014), who presented a methodology for the identification and quantification of the non-linear contribution to the interaction information, able to identify main sources of nonlinearity in the nodes couplings. A comparison of the spatial structures uncovered in Fig. 1(b) here, with those in Hlinka et al. (2014), Fig. 5 (central panel), reveals that some of the areas with large distance values in Fig. 1(b) tend to coincide with the areas with nonlinear contribution to the mutual information. For example, the African and the Amazon rainforests are clearly seen in both maps. However, there are also differences, for example, a ring in the Ocean around the Antarctica and a well defined region near the north pole (Greenland sea) that are strong signals in Fig. 5 of Hlinka et al. (2014) but are not strong in Fig. 1(b) here. However, in Fig. 3(b) here, we note that these regions are regions of low SAT anomaly entropy (due to the low temperature tail of SAT anomaly pdf). In other words, the regions with nonlinear contribution to mutual information seen in Fig. 5 of Hlinka



et al. (2014) are seen either in Fig. 1(b) or in Fig. 3(b) here. These observations suggest that the connectivity of these regions might reflect similar response to solar forcing, and/or similar SAT variability, with long-tailed distribution of SAT anomalies.

## 5 Conclusions

In this work we have investigated statistical properties of a climatological field (the surface air temperature, SAT) using two monthly reanalysis datasets: ERA Interim and NCEP CDAS1. We have quantified SAT stochasticity by means of Shannon entropy, $H_i$, and we have analyzed the response to solar forcing, by means of a distance measure, $d_i$, that assesses the similarity in the shape of the time series of the top-of-atmosphere incoming solar radiation (the insolation, $x_i$) and the SAT, $y_i$, both having been previously normalized to have zero mean and unit variance. A delay in the response of SAT to solar forcing was taken into account by lagging SAT with respect to the insolation.

We found that these two measures provide meaningful insight into global properties of SAT time series. In the distance map, Fig. 1(b), we found that the tropics show considerable larger distance values, in comparison with the extratropics. There are well-defined structures in the oceans, over the Intertropical Convergence Zones, and over some continental areas, especially in regions largely dominated by monsoons, such as the tropical rainforest in Africa and South America, as well as over India. These regions do not respond to local insolation, but are characterized by strong air-sea coupling or land-atmosphere interaction which involve non-local processes.

In the SAT entropy maps we found very similar spatial structures, but with opposite trend, in the sense that the geographical regions with high $d_i$ values correspond, in general, to regions with low entropy. When the entropy was calculated from the SAT anomalies, the tropical spatial patterns disappeared but those in the high latitudes remained. This was interpreted as due to the fact that in high latitudes, mainly because of the presence of sea ice, there is a strong seasonality in variance that remains even when the annual cycle is removed.

The entropy analysis also allowed to identify, in a well-defined region of the tropical western Pacific, a remarkable difference between the ERA Interim and the NCEP CDAS1 data sets, which is due to the presence of extreme values in one dataset but not in the other. Therefore, entropy maps can be a valuable tool for anomaly detection and model inter-comparisons.

As future work, it will be interesting to perform a similar analysis using the CMIP5 database to determine how the spatial patterns and lag times identified here may change under an scenario of climate change.

*Acknowledgements.* This work was supported by the LINC project (FP7-PEOPLE-2011-ITN, Grant No. 289447). C. M. also acknowledges partial support from Spanish MINECO (FIS2015-66503-C3-2-P) and ICREA ACADEMIA.





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
