# Peer review of "Identifying global patterns of stochasticity and nonlinearity in the Earth System"

_Earth System Dynamics, 2016_

## Referee Comment (RC1) · Anonymous Referee #1 · 10 Nov 2016

While the topic is potentially interesting for the earth system dynamics community, the manuscript is my opinion is not suitable for publication. The paper applies two known metrics to compare temperature and radiation time series from different locations, in search for nonlinear interactions and deterministic vs random components. Interactions are obviously expected in such time series and the results are presented di forms of spatial maps, without attempt to go in depth into the physical, meteorological and climatic meaning of the findings. Very little attention is given to the underlying physics or to the meaning of nonlinearity and stochasticity's (time scales? Determinism?) for the underlying radiative-forced climate system.

For these reasons, the paper does not represent a novel enough contribution to the Geophysical literature to be recommended for publication in ESD.

---

## Referee Comment (RC2) · Anonymous Referee #2 · 12 Nov 2016

In the article "Identifying global patterns of stochasticity and nonlinearity in the Earth System", the authors use two metrics to characterize the properties of air surface temperatures (SAT) in the ERA and the NCEP reanalyses. The authors claim that the distance between the lagged SAT time series and the insolation allows to quantify whether the insolation is the main responsible of local SAT variations. They also suggest that the Shannon entropy is a measure of stochasticity of the SAT time series.

I like the idea of using dynamical metrics to undercover properties of the climate system but I find that the work by Arizmendi et al. does not provide enough elements to support the claims of the authors. I will try to highlight the problems of the manuscript on different levels. I hope that the authors will consider my suggestions to rethink/rewrite their work that, in my opinion, should not be further considered for publication in ESD.

[Figure]

1) Methodological problems: everywhere in the manuscript there is confusion between: forcing, physical processes, internal variability, turbulence-uncertainty-stochasticity. I suggest the author to revise the book: "Chaos And Turbulence: An Introduction To Nonlinear Dynamics And Complex Systems" by Paul Manneville, which explain most of those concepts. In particular they write in the introduction that: "... because of the physical processes that govern our climate (ocean and atmospheric processes, solar forcing, vegetation, human activity, etc.)". In my view, ocean and atmospheric processes are physical processes of the climate system while solar forcing is an external driver that sets the climate system out of equilibrium. Another important point is about the Shannon entropy: I strongly disagree that entropy is a measure of "stochasticity". The adjective stochastic refers to the random behavior of a system which is not the same as "disordered" in the sense of Shannon. Well-known examples are the class of deterministic chaotic dynamical systems (Lorenz 1963, Rossler, Henon, ...) that have a certain degree of disorder although they have no stochastic components. If the authors really wants to test the stochasticity of the SAT time series, I strongly reccomend to use the results by Rosso, O. A., et al. "Distinguishing noise from chaos." Physical review letters99.15 (2007): 154102. Last but not least, the stochastic behavior observed by the authros could also be due to turbulence: turbulence is different from noise, as the authors know for sure. The authors never comment on the role of turbulence while, especially at the tropics, turbulence is an important player for the atmospheric dynamics.

2) Organization/interpretation of the results: The link between PDF shape, extreme values and Shannon entropy is not shown/explained. The authors make a list of processes that, when the insolation is not linked to the local SAT, should be responsible for the variations of SAT, but they do not provide any analysis or physical justifications of their speculations. In the manuscript, it is claimed that Shannon entropy is good at distinguish ERA from NCEP reanalysis but how this compares to simpler statistical metrics like the difference in the local SAT means, variance, skewness, . . .? Why the authors do not provide maps of differences between ERA and NCEP reanalyses? Is it because the datasets are at different resolutions? If the answer is yes, how do

they consider the impact of different grid-sizes on the results? The authors normalize the distances by subtracting the mean and dividing by the variance and claim that this removes all the memory effects. This is false: it removes memory effects up to the second order, but there could be higher order memory effects still hidden in the datasets. Finally, the supplementary information is too short to justify the need for a separate document. I suggest either to expand (for example by comparing the authors' metrics with simpler statistical metrics) or integrate the results in the main text.

---

## Referee Comment (RC3) · Anonymous Referee #3 · 13 Nov 2016

In the manuscript "Identifying global patterns of stochasticity and nonlinearity in the Earth System" by Arizmendi et al. the authors adopt two metrics from information theory to characterize the properties of surface air temperature (SAT) and the relationship between SAT and solar forcing. While I strongly support the use of methods from information theory and neural network to investigate climate patterns and relationships, I find the manuscript in its current form not suitable for publication in ESD. The interpretation of the results is very superficial or absent, and the authors do not provide any physical justification for their findings. The authors mention various climate processes that could help explain Figure 1 and 2, but no attempt is made to an attribution based on physical principles/mechanisms except for unsubstantiated speculations. It is not investigated in any detail why ERA and NCEP are so different in the warm pool (i.e. are the extreme values responsible for the differences reasonable or not and what is

the mechanism by which those values appear?). It is not discussed what role does resolution play in the analysis presented and more generally how Shannon entrophy depends on resolution. Figure 3 has orange squares underneath the black circles and its unclear how meaningful is overall. Finally, it is not obvious how the tail of the PDF in figure 4 and the Shannon entropy are linked (I find Fig 4 very confusing but instead it should explain the differences between the two reanalyses in Figure 2)

The lack of any in depth interpretation adds to confusing statements about stochasticity repeated throughout the manuscript. Shannon entropy does not quantify the degree of stochasticity, but inform on the degree of unpredictability of a signal. It is not obvious or proven that extreme values of Shannon entropy for SAT are stochastic or due to stochastic processes (chaos does not have to be stochastic).

There are few typos throughout the manuscript.

---

## Short Comment (SC1) · 21 Nov 2016

Taking into account the referees' comments, we would like to withdrawn our manuscript. While we disagree with the referees' opinion, we thank the three referees for their comments that will allow us to improve our work, which will be submitted to another journal. We present below our response to the specific comments.

Response to Referee #1 While the metrics that we use are known, they have not yet been applied to investigate long-range climate patterns or to compare climate datasets. The novel contribution of our work to the climate literature is the demonstration that these metrics can be very useful for examining and comparing climate datasets.

[Figure]

---

## Short Comment (SC2) · 21 Nov 2016

Taking into account the referees' comments, we would like to withdrawn our manuscript. While we disagree with the referees' opinion, we thank the three referees for their comments that will allow us to improve our work, which will be submitted to another journal. We present below our response to the specific comments.

Response to Referee #2 We agree with the referee that "the adjective stochastic refers to the random behavior of a system..." and we also agree with the referee that "is well-known that deterministic chaotic dynamical systems can have a certain degree of disorder although they have no stochastic components." To explain what we mean by "stochasticity", in the introduction we say that the entropy allows to "...quantify the "noisy" nature of SAT variability (to which we refer to as stochasticity)..." We agree

that this idea can explained more clearly by rephrasing this sentence as "...quantify the degree of unpredictability of SAT variability" as indicated by Referee #3.

The referee says that "The authors normalize the distances by subtracting the mean and dividing by the variance and claim that this removes all the memory effects."

This is not precise: we do not claim that this removes all memory effects. In the manuscript we say that "Here tau_i is a lag that takes into account inertia and/or memory effects." Moreover, in the introduction we discuss the fact that more general forms could include more than one lag time. Nevertheless, it is remarkable that such a simple expression allows uncovering meaningful long-range climate patterns.

———————————————————

---

## Short Comment (SC3) · 21 Nov 2016

Taking into account the referees' comments, we would like to withdrawn our manuscript. While we disagree with the referees' opinion, we thank the three referees for their comments that will allow us to improve our work, which will be submitted to another journal. We present below our response to the specific comments.

Response to Referee #3 We remark that the relation between Shannon entropy (not entrophy) and the long tail shape of the PDF should be sufficiently clear for anyone that is familiar with the concept of Shannon entropy: the entropy is maximum if the PDF is uniform and is minimum if the PDF is a delta. We have performed additional calculations with the NCEP CDAS1 dataset and found that the resolution (10/20/40 bins) does not significantly modify the entropy maps. The reviewer says "It is not investigated in

[Figure]

any detail why ERA and NCEP are so different in the warm pool (i.e. are the extreme values responsible for the differences reasonable or not and what is the mechanism by which those values appear?)".

We don't know why the two datasets are different in the warm pool and we also don't know if the extreme values are reasonable or not. However, we remark that this difference is a very relevant result of our analysis, and the scientific community should be aware of this difference in this region. We agree with the reviewer that Fig. 4, aimed at explaining the differences, is not sufficiently clear and will be redone.

The reviewer says "Figure 3 has orange squares underneath the black circles and its unclear how meaningful is overall." Of course the separation is not 100% but the bars that indicate the inter-quartile ranges clearly demonstrate the trend: higher entropy – lower distance.

The reviewer says "It is not obvious or proven that extreme values of Shannon entrophy for SAT are stochastic or due to stochastic processes". We fully agree and we don't say or mean that idea in the manuscript.
* * *

---

## Author Comment (AC1) · 7 Dec 2016

Taking into account the referees' comments, we **would like to withdrawn our manuscript**. While we disagree with the referees' opinion, we thank the three referees for their comments that will allow us to improve our work, which will be submitted to another journal. We present below our response to the specific comments.

**Response to Referee #1**

While the metrics that we use are known, they have not yet been applied to investigate long-range climate patterns or to compare climate datasets. The novel contribution of our work to the climate literature is the demonstration that these metrics can be very useful for examining and comparing climate datasets.

**Response to Referee #2**

We agree with the referee that "the adjective stochastic refers to the random behavior of a system…" and we also agree with the referee that "is well-known that deterministic chaotic dynamical systems can have a certain degree of disorder although they have no stochastic components." To explain what we mean by "stochasticity", in the introduction we say that the entropy allows to "…quantify the "noisy" nature of SAT variability (to which we refer to as stochasticity)…" We agree that this idea can explained more clearly by rephrasing this sentence as "…quantify the *degree of unpredictability* of SAT variability" as indicated by Referee #3.

The referee says that "The authors normalize the distances by subtracting the mean and dividing by the variance and claim that this removes all the memory effects."

This is not precise: we do not claim that this removes all memory effects. In the manuscript we say that "Here tau_i is a lag that takes into account inertia and/or memory effects." Moreover, in the introduction we discuss the fact that more general forms could include more than one lag time. Nevertheless, it is remarkable that such a simple expression allows uncovering meaningful long-range climate patterns.

**Response to Referee #3**

We remark that the relation between Shannon entropy (not entrophy) and the long tail shape of the PDF should be sufficiently clear for anyone that is familiar with the concept of Shannon entropy: the entropy is maximum if the PDF is uniform and is minimum if the PDF is a delta. We have performed additional calculations with the NCEP CDAS1 dataset and found that the resolution (10/20/40 bins) does not significantly modify the entropy maps.

The reviewer says "It is not investigated in any detail why ERA and NCEP are so different in the warm pool (i.e. are the extreme values responsible for the differences reasonable or not and what is the mechanism by which those values appear?)".

We don't know why the two datasets are different in the warm pool and we also don't know if the extreme values are reasonable or not. However, we remark that this difference is a very relevant result of our analysis, and the scientific community should be aware of this difference in this region. We agree with the reviewer that Fig. 4, aimed at explaining the differences, is not sufficiently clear and will be redone.

The reviewer says "Figure 3 has orange squares underneath the black circles and its unclear how meaningful is overall." Of course the separation is not 100% but the bars that indicate the inter-quartile ranges clearly demonstrate the trend: higher entropy – lower distance.

The reviewer says "It is not obvious or proven that extreme values of Shannon entropy for SAT are stochastic or due to stochastic processes". We fully agree and we don't say or mean that idea in the manuscript.